# The impact of the digital economy on creative industries development: Empirical evidence based on the China

Xiaodi Zhao[1], Lei Shen[1]*, Zhengyun Jiang[2]

1 Glorious Sun School of Business and Management, Donghua University, Shanghai, China, 2 Faculty of Geographical Science, Beijing Normal University, Beijing, China

* slei@dhu.edu.cn

**Data Availability Statement:** All relevant data are within the manuscript and its Supporting Information files.

## Abstract

Digital economy has become a "new engine" that driving global economic growth. Nevertheless, numerous controversies persist regarding whether and how digital economy can facilitate the development of emerging industries. Thus, this paper investigates how digital economy affects creative industries development in China and whether innovation efficiency mediates this relationship. Drawing upon a panel data set containing 29 Chinese provinces from 2012 to 2019, an econometric model is constructed for empirical analysis. We find that digital economy significantly promotes creative industries development, and innovation efficiency plays a partial mediating role between digital economy and creative industries development. According to the influence mechanism, the digital economy of various regions could promote the creative industries development by improving the innovation efficiency. Finally, relevant suggestions were put forward from the expanding application paths, improving regional innovation efficiency, and creating an innovative environment.

## 1. Introduction

In recent years, the advances in digital technologies such as big data, cloud computing and block-chain have provided new impetus and direction for economic transformation and development, and contributed to the optimization and upgrading of the industrial structure. With the continuous integration and development of digital technology and its continuous penetration into social and economic forms, the digital economy has emerged as the times require, and has become the most potential and dynamic new economic form. The World Economic Forum has defined the digital economy as "With digital knowledge and information as key production factors, modern information networks as important carriers, and information and communication technology as an essential component of increasing productivity through economic activities". Nowadays, the digital economy represents an emerging industrial revolution that is reliant on high-tech digital technology and rapidly expanded globally [1]. According to the "China Digital Economy Development Report (2022)" released by the China Academy of Information and Communications Technology (CAICT), China's digital economy made a new breakthrough in 2021, with the total digital economy reaching 45.5 trillion yuan,

**Funding:** The authors received no specific funding for this work.

**Competing interests:** The authors have declared that no competing interests exist.

3.4percentage points higher than the nominal growth rate of GDP in the same period, and accounted for 39.8% of GDP. The supporting role of the digital economy in the national economy has become more obvious, and its impact on the economy of other industries in society has also attracted widespread attention from academia.

Creative industries are growing up in the context of economic globalization. It is an emerging industry with innovative creativity as its core. In the late 1980s and 1990s, several industrial countries governments began to consider creativity as a contributor to the national economy, looking to deal with globalization [2], different literature have also defined the concept of creative industries [3]. Over time, the creative industries have become an important engine of economic growth, job creation, social cohesion [4], and an important component of the global economy.

The digital economy and the creative industries are both part of Global Economic Transformation and Development and have been the focus of academic research and government intervention [2]. The digital economy has provided innovative approaches and solutions for global economies, particularly those in transition, to address societal challenges arising from the previous overemphasis on rapid economic growth, such as distorted industrial structures, severe environmental pollution, and regional imbalances, and has facilitated the transformation of the economic development model from rapid expansion to high-quality growth. Meanwhile, given its high value-added and innovative nature, the creative industry emerges as a pivotal sector for achieving environmentally sustainable and green-oriented economic development. Therefore, it holds great significance to explore the role of the digital economy in promoting the creative industry in order to realize the goal of green development.

Currently, the extensive research on the digital economy has mainly focused on three main areas: First, existing studies focus on its impact on economic growth, and it promotes the development of all aspects of the social economy with information and communication technology as the core technological tools [5], and alleviate the congestion effect in the advanced areas, increasing entrepreneurial activity and enhance energy intensity, thus providing a great foundation for economic development [6, 7]. Second, the studies of the digital economy on the environmental field and green development, such as promoting the clean energy development [8], Optimizing Urban Carbon Emissions [9], Control of urban environmental pollution, etc. [10]. The third aspect is the impact on other industries, such as tourism [11–13], development of manufacturing industry [14], the business model of creative industry, etc. [15]. For the creative industry, it is considered to be the center of new organizational and business practices [16], and a significant sector of the world economy [17, 18]. But there is no standardized definition and designation for creative industries, the UK adopts the title of creative industry, the US defines it as copyright industry, the Spain defines it as cultural and leisure industry, and the Japan defines it as content industry. It can be recognized that creative industries have obvious cross-industry characteristics. Meanwhile, influenced by the development of digital technology, the business models of creative industries are undergoing profound changes, some business models derived from digital technology are being born (e.g. games), traditional creative industry sectors are being transformed by digital technology (e.g. publishing, advertising, design and music), and the influence of digital technology in some creative industry fields is gradually deepening (e.g. museums, fine arts, cultural heritage, etc.) [15]. Existing research has focused on the characteristics of their business models and on characterizing the effect of creative industries in economic development. However, little is known about how to measure the level of development of regional creative industries and how the digital economy affects the development of creative industries.

The creative industries are centered on innovation and creativity, emphasizing culture, technology, innovation, creativity and intellectual property, its sustainable development

requires technological change and accumulation of creative elements. And the development of the digital economy is accompanied by the application, development and popularization of digital technologies such as cloud computing, the internet and the internet of Things. which has driven a wide flow of technology, talent and capital, leading to a new round of technological innovation [19], has significantly improved the level of regional innovation efficiency in China, and has become a new driving force for the country to improve regional innovation efficiency in the new era [20], then brings the gathering of creative elements and provides a constant impetus for the creative industries. Therefore, we need to consider the following questions: whether the continuous development of regional innovation activities is influenced by the development of the digital economy and whether the efficiency of regional innovation can enhance the development of creative industries? The existing studies have only partially discussed the above issues, however, there is no clear answer on whether innovation efficiency mediates the relationship between digital economy and creative industries development.

Thus, there are three major gaps in the existing literature. First, previous studies mostly focus on how the digital economy has affected economic growth, green development, business model, and other issues. They have failed to find a causal relationship between the digital economy and creative industries. But as a new industry, the development of creative industries cannot be separated from the progress of technology and the accumulation of creative elements. And whether the technology, human resources and innovative elements brought by the digital economy can promote the development of creative industries is an important issue to be considered. Second, due to the diverse concepts of creative industries and their cross-industry characteristics, there is no unified standard for measuring the degree of development of creative industries. Third, the effect mechanism of digital economy on creative elements is still poorly understood. Hence, to make the promotion of the digital economy on the development of creative industries more obvious and to propose policy recommendations to promote the development of creative industries, it would be of great benefit to examine the influencing mechanism.

Therefore, this paper focuses on two research questions. The aim is to investigate the relationship between digital economy and creative industries and validating the mediating effect of innovation efficiency on them. If the mediating effect of innovation efficiency can be empirically validated, then it is theoretically possible to answer the important question of how the digital economy promote the development of creative industries.

Based on the above considerations, the marginal contributions of this study include further improvement in the academic understanding of the relationship between the digital economy and creative industries development. Specifically, based on comprehensive internal and external considerations and data accessibility, we construct a comprehensive evaluation index to measure the creative industries and digital economy on 29 provinces in China from 2012 to 2019. Digital economy, innovation efficiency and creative industries development are integrated into a unified analytical framework to elucidate the complex logical relationships among them. And the two-way fixed effects model is applied to investigate the effect between the digital economy and creative industries, the several robustness tests were applied to validate results. Compare with existing research, this paper is one of the first efforts to examine the relationship between digital economy and creative industries development.

## 2. Literature review

As significant catalysts for green economic development, digital economy and creative industries have garnered considerable research attention. The digital economy is closely related to economic development, environmental protection, and corporate governance. With the

growing role of information and communications technology in all sectors of the economy, more and more experts are beginning to test the relationship between the digital economy and economic development [21]. Existing studies indicate that the digital economy has a positive impact on economic growth [6, 22, 23]. Essentially, the development of the digital economy is accompanied by advances in communications infrastructure that ensure faster access to information and stimulate innovation and knowledge sharing, leading to economic progress [21]. The technological innovation and industrial structure adjustment are important ways for digital economy to improve the efficiency level of green economy [1, 24]. In terms of environmental protection, there is a non-linear relationship between the digital economy and carbon emissions. Some studies have shown that the digital economy has slowed down carbon emissions, reduced air pollution, and promoted green development of the economy [8–10, 25, 26], But there are also studies indicating that the digital economy will increase energy consumption [27]. At the micro level, digital technologies have made a substantial significant contribution to firm output [28], force financial services institutions to adopt technologies that help deliver high-quality services at a minimal cost [29]. Digital Economy also provides a driving force for the upgrading and sustainable development of industries such as tourism and manufacturing [12, 14, 30].

The notion of creative industries emerged in the 1990s, it was proposed by the government to address the development of film, media, publishing, video games, design and interactive entertainment [16]. Scholars have studied creative industries from various perspectives, including social market, government support, and technological progress. Despite the fact that the creative industries are significant and growing rapidly over the past years [31], but the terms "cultural industries" and "creative industries" are often confused, there is no formal explanation of the difference between them [32]. To distinguish the terms, The UK's Department for Digital, Culture, Media and Sport (DCMS) definition of the creative industries as" those industries which have their origin in individual creativity, skill, and talent and which have a potential for wealth and job creation through the generation and exploitation of intellectual property" [33]. This definition has become a de facto world standard in recent years [31]. Some scholars also disagree with the definition of creative industries and believe that creative economy should be redefined under different applicable circumstances [15, 16, 32, 34, 35]. As an important part of the knowledge economy, the creative industries can be characterized as entrepreneurial, innovative, sustainable, and flexible [3], and they have a broad reach in both traditional (e.g., arts, publishing) and digitally native sectors (e.g., video games) [15]. It is believed that creative industries are the hub of new organizations and business practices, and digital technology will bring changes in business models for creative industries, while radical innovation and incremental innovation of creative enterprises are closely related to government policies. From the above analysis, it can be concluded that the creative industries have significant cross-industry characteristics, making it difficult to measure its level of development. And the measurement of it in existing research results is often confused with the cultural industry and cannot accurately reflect the level of development of the creative industries [36].

Current research on the relationship between the digital economy and the creative industries focuses on two main areas. The first is on the impact of digital technologies on the creative industries, for example, the block-chain based non-fungible tokens and smart contracts offer exciting opportunities for the creative industries [37], and digital technologies can facilitate business model innovations in the creative industries, etc. Secondly, regarding the relationship between the digital economy and the creative industries, some scholars believe that the creative economy needs to rely on the digital economy, while the digital economy also needs creative input, and the two are reliant on each other [2]. Unfortunately, there is a relative lack of previous literature correlating the digital economy with creative industries. The effects and potential

mechanisms of digital economy in creative industries development can only be inferred from other relevant studies.

Based on the above, the reason why the digital economy can have an impact on economic development, environmental optimization, and other industrial development is mainly based on its two characteristics. The first is technical attribute: the continuous development of digital technology has improved production efficiency, reduced production costs, and promoted the transformation and upgrading of traditional industries towards digitization, automation, and intelligence [38], thereby promoting the economic development of enterprises and driving regional economic development. The second is the innovation attribute: The development of digital economy implies the construction and improvement of communication and Internet infrastructure, which breaks the spatial limitation of information and knowledge, accelerates the dissemination of knowledge and technology within the region, reduces the limitation of time and space on innovation activities, promotes the flow of innovation factors and improves innovation efficiency in the region, thus contributing to the creation of a regional innovation environment [39, 40]. For creative industries, their development requires the support of technology and creative elements. Therefore, from a theoretical point of view, the innovation efficiency is closely related to the development of the digital economy and is also the main driving factor for the development of the creative industry. So, whether it can be a mediating variable between the digital economy and creative industries is one of the key issues to be discussed in academia.

In sum, this research aims to make the following contributions to the literature. Firstly, this study constructs a comprehensive index system for measuring creative industries at the province level, and bridges the gap in quantitative research about creative industries. Secondly, this study innovatively investigates the causal relationship between the digital economy and creative industries, bringing a new perspective for researchers studying the limited but emerging literature. Finally, this paper contributes to a more comprehensive understanding of the mechanisms through which the digital economy affects creative industries by investigating the mediating roles of innovation efficiency.

## 3. Theoretical framework and hypotheses

This study demonstrates the influential mechanisms of the digital economy on creative industries development. Fig 1 depicts our conceptual framework and hypotheses.

### 3.1 Digital economy and creative industries development

Digital economy is a series of economic activities based on digital technology, digital platform as the main medium, and digital enabling infrastructure as an important support [41]. These economic activities are dynamics, and have profound impacts on all areas of the national

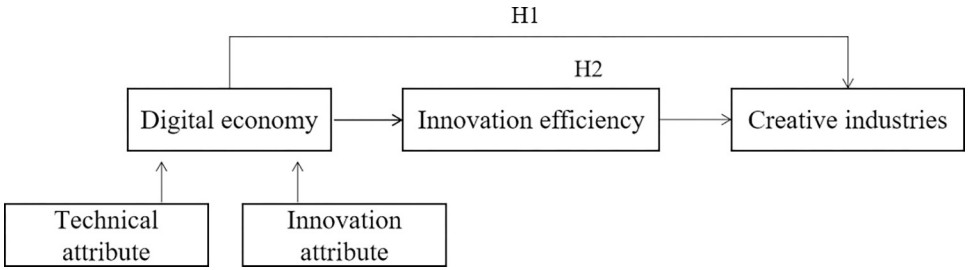

**Fig 1. The conceptual framework of the digital economy on creative industries development.**

economy, such as the productivity enhancement in traditional industries, the market efficiency effects, and the potential arising from entirely new products and industries [42]. On a micro level, digital transformation has also become a way for companies to attain competitive advantage and differentiation [43]. Based on the above analysis, the digital economy promotes the development of creative industries mainly from the following aspects: First, Digital economy promotes product and business model innovation in creative industries. Through the deep integration of digital technology and creative industries, it promotes product innovation and enriches the form of creative industries. Second, the digital economy has enriched the creative industry marketing model. The development of the digital economy has been accompanied by the improvement of communication and Internet settings, allowing direct dialogue between consumers and suppliers, pulling consumer demand through precise marketing and reducing the constraints of time and space on industrial development. Third, the digital economy optimizes the development environment of creative industries. The digital economy alleviates information asymmetry, accelerates the flow of creative elements, optimizes the allocation of capital, human and creative resources, and injects new vitality into the development of creative enterprises. Focusing on the driving role of digital technology in the creative industries development can further embolden the demand for digital transformation and the development of creative resources; therefore, we propose:

Hypothesis 1: Digital economy is conducive to the improvement of creative industries development.

## 3.2 Mediating effect of innovation efficiency on digital economy and creative industries development

In the process of digital economy's impact on creative industries, the improvement of innovation efficiency is an important product of digital economy development and a major driver to enhance the development of creative industries.

At the macro level, the application of digital technology has changed the nature and structure of new products and services, spawned novel value creation and value appropriation pathways, and changed the entire industry in a broader sense [40], and improved the innovation environment. Meanwhile, the development of digital economy has driven the concentration of high-end talents, high-tech enterprises, R&D capital and other innovation elements, has generated new economic growth points to the society, achieving the transformation from factor-driven to innovation-driven, and improving the innovation efficiency of the city. At the micro-level, Digital transformation can prompt enterprises to conduct business model innovation, drive enterprises to improve production technology [44]. Meanwhile, the development of digital technologies can help reduce enterprises' costs of carrying out innovation [45].

From the perspective of creative industries, innovation and creation are the core competitiveness of their development. The improvement of innovation efficiency implies the concentration of technical level, creative elements, human capital and other elements, which provides a good innovation environment for the development of creative industries and thus improves their sustainable development. Based on these analyses, we propose the second hypothesis:

Hypothesis 2: Innovation efficiency mediates the relationship between digital economy and creative industries development

## 4. Methodology

### 4.1 Explanation of variables

**4.1.1 Explained variable.** In terms of creative industries development, there is no uniform standard for the measurement of creative industries in existing research. Referring to the

definition of the UK's Department for Digital, Culture, Media and Sport (DCMS) of the creative industries as "those industries which have their origin in individual creativity, skill, and talent and which have a potential for wealth and job creation through the generation and exploitation of intellectual property" [33]. Based on this definition, DCMS classifies creative industries, including advertising and marketing; architecture; design and designer fashion; film, TV, video, radio and photography; IT, software and computer service; music, performing and visual arts; publishing. With reference to the above definitions and classifications and considering the accessibility of data, we have constructed a measurement system for the development of China's creative industries, as shown Table 1.

According to the above analysis and the construction of creative industries indicator system, this paper adopted the entropy method to calculate the development index of regional creative industries.

**4.1.2 Core explanatory variable.** This paper adopted the development level of the digital economy as the core explanatory variable. The digital economy, in essence, is the result of the deep integration between the traditional economy and digital technologies [46]. It represents a series of economic activities that are based on digital technology, primarily mediated by digital platforms, and supported by digital empowerment infrastructure [41]. The measurement system of digital economy development no uniform standard, but it has been continuously studied by scholars. Referring to other scholars' research on digital economy measurement [6, 9, 13], we investigate the digital economy from four dimensions: Digital Infrastructure, Digital Industry Development, Digital Innovation Capability and Digital finance. At the same time, proxy variables are selected for the above dimensions. Among them, digital infrastructure refers to the digital technology-enabled software, communication facilities, computer hardware, and other components that serve as the foundation for ensuring the operation and development of the digital economy. Therefore, referring to existing research [13], digital infrastructure is represented by internet penetration rate and mobile communication coverage rate, and measured by the number of Internet users per 100 people and the number of cell phones users per 100 people. The development of the digital industry mainly considers the

**Table 1. Construction of creative industries indicator system.**

| Target index | Secondary indexes | Three-level indexes |
|---|---|---|
| creative industries | advertising and marketing | Regional advertising business volume |
| | architecture | Number of Survey and design firm |
| | | Number of employees in survey and design firms |
| | design and designer fashion | Number of design-based patent applications granted |
| | Film, TV, video, radio and photography | Integrated population coverage of radio programs |
| | | Integrated population coverage of TV programs |
| | | Total Radio & TV Revenue |
| | | Broadcast of self-produced radio programs |
| | | Broadcasting of self-produced TV programs |
| | IT, software and computer service | Software business revenues |
| | publishing | Basic information of publications |
| | | Registration of copyright contracts |
| | music, performing and visual arts | Regional cultural system art performance revenue |
| | | Registration of Artworks (Writing, music, opera, dance, art, photography, film and television) |
| | | Number of institutions of cultural social organizations |
| | | Regional financial expenditure on culture, sports and media |

**Table 2. Construction of digital economy indicator system.**

| Target index | Secondary indexes | Three-level indexes |
|---|---|---|
| Digital infrastructure | internet penetration rate | the number of Internet users per 100 people |
| | mobile communication coverage rate | the number of cell phones users per 100 people |
| Digital industry development | Internet industry output | the telecommunications business per 10,000 people |
| | information industry development | percentage of employees in information transmission and software and information technology services |
| | e-commerce business development | postal revenue per capital |
| Digital innovation capability | digital innovation element support | government science and technology expenditure |
| | digital High-Tech Penetration | the degree of penetration of digital high-tech applications in listed companies |
| Digital finance | breadth of coverage | Digital Financial Inclusion Index |
| | depth of use | Digital Financial Inclusion Index |
| | digitization of digital finance | Digital Financial Inclusion Index |

level of development of core digital technology industries. Draw on existing research findings [9, 13, 46], digital industry development is represented by internet industry output, information industry development and e-commerce business development, and measured by the telecommunications business per 10,000 people, percentage of employees in information transmission and software and information technology services, postal revenue per capital. Digital innovation capacity refers to the potential for digital technology-driven regional innovation and development, mainly represented by digital innovation element support and digital high-tech penetration, and measured by government science and technology expenditure and the degree of penetration of digital high-tech applications in listed companies. Digital finance is represented by the digital financial inclusion index (DFII), compiled by Peking University and Ant Financial jointly, which is measured by the breadth of coverage, depth of use, and digitization of digital finance [47]. Then, the measurement system of digital economy development in China was constructed as shown in the Table 2.

**4.1.3 Mediating variable.** Based on the above analysis, this paper adopted innovation efficiency (IN) as the mediating mechanism of the digital economy affecting creative industries. and referring other studies [48], the innovation efficiency is represented by the number of patents granted per 10,000 people.

**4.1.4 Control variables.** Due to the wide variety of factors that contribute to the development of creative industries, with reference to existing studies, the following control variables were added to the model: (1) economic development (ED), measured by per capital GDP; (2) industrial structure upgrading (IS): measured by the ratio of tertiary industry to secondary industry output value. a larger ratio indicates a more optimized industrial structure; (3) level of opening up (OP), measured by the ratio of the foreign direct investment (FDI) to GDP. and (4) governmental support (GS), measured by the ratio of government spending on culture, sports and media to total spending.

### 4.2 Model construction

**4.2.1 Direct effect model.** To test the direct effect of digital economy on creative industries development, combining the theoretical mechanisms analyzed, the following regression model was adopted:

$$cre_{it} = \alpha_0 + \beta_1 Dig_{it} + \beta_2 Controls_{it} + \gamma_i + \sigma_t + \varepsilon_{it} \tag{1}$$

Where $t$ represents the year; $i$ represents the province; *Cre* represents the level of creative industries development; *Dig* represents the level of digital economy development; *Controls* represents a series of control variables; $\beta$ represents the regression coefficients of the effects of each explanatory variable on the development of creative industries; $\gamma$ represents region fixed effect; $\sigma$ represents time fixed effect, and $\varepsilon_{it}$ represents a random error term.

**4.2.2 Mediating effect model.** To examine hypothesis 2, with reference to Baron and Kenny [49], this paper utilized mediating effect model, as follows:

$$cre_{it} = C + \alpha Dig_{it} + \gamma Controls_{it} + \gamma_i + \delta_t + \mu_{it} \tag{2}$$

$$In_{it} = C + \eta Dig_{it} + \omega Controls_{it} + \gamma_i + \delta_t + \mu_{it} \tag{3}$$

$$cre_{it} = C + \theta Dig_{it} + \lambda In_{it} + \varepsilon Controls_{it} + \gamma_i + \delta_t + \mu_{it} \tag{4}$$

Where $In_{it}$ represents the mediating variable. $\alpha$ represents the impact effect of digital economy on creative industries. $\omega$ represents the impact effect of digital economy on innovation efficiency. $\theta$ represents the direct impact of the digital economy on the creative industries, and $\lambda$ represents the impact of innovation efficiency on the creative industries based on controlling for the effect of the digital economy on the creative industries. If the coefficient $\alpha$ is significant, and both $\eta$ and $\lambda$ are significant, the existence of a mediating effect is confirmed; If the coefficient $\theta$ is significant, there is a partial mediating effect; if $\theta$ is not significant, there is a complete mediation effect.

## 4.3 Data description

The empirical analysis was taken from on a panel data set of the 29 Chinese provinces for the period 2012–2019. Considering the availability of data, our study does not cover Hong Kong, Macao, Taiwan, Xinjiang and Tibet. In addition, there are two reasons for taking data from 2012 to 2019: Firstly, China's digital technology has grown rapidly in 2011. And the specification of creative industries relevant statistics mainly started in 2012. At the same time, creative industries are greatly affected by the COVID-19, so the data of creative industries after 2019 are not applicable to this study.

The explained variable of this study are creative industries, The data comes mostly from the China Statistical Yearbook of Culture and Related Industries, the official websites of provincial governments and statistical bureaus. The explanatory variables and control variables are mainly obtained from China City Statistical Yearbook, EPS Database, China Statistical Yearbook and statistical yearbooks of each province. The digital financial inclusion index is measured using the China Digital Financial Inclusion Index released by the Digital Finance Research Center of Peking University in cooperation with Ant Financial Services Group [47]. In particular, referring to other scholars' practices, the digital high-tech penetration mainly calculated the keywords of integrated circuit, internet of Things, big data, artificial intelligence, cloud computing, block chain, ICT industry, smart city, mobile Internet, data mining, digital trade, mobile payment, satellite navigation, electronic commerce, internet plus, data visualization, virtual reality and other keywords appearing in the business scope of listed companies to reflect the penetration level of high technology, and then aggregated to the provincial scale. Due to the wide variety of indicators involved, all indicators are dimensionless in order to eliminate the influence of the indicators on the evaluation results caused by the different units of measurement.

## 5. Empirical results

### 5.1 Baseline regression results

Before estimating the model, we conducted a Hausman test using Stata 17 software to determine whether to use a random effects model or a fixed effects model in this paper. The results of the Hausman test indicated that the fixed effects model is more suitable for the research design. Subsequently, within the fixed effects model, we considered time effects. To examine the joint significance of all yearly dummy variables, we conducted a test and found that the p-value was less than 0.01. Therefore, we strongly rejected the hypothesis of "no time effects" and concluded that the model includes time effects. Then, this paper uses a two-way fixed effects model that considers both time and individual effects. and clustering robust standard errors to eliminate the effect of heteroskedasticity on the model.

Table 3 presents the regression results of the fixed effects model. In column (1) of Table 3, we only investigate the impact of digital economy on creative industries development in Chinese provinces. In column (2) of Table 3, we include both the control variables and the indicator of digital economy in the regression model. The $R^2$ values of both regression models were above 0.9, indicating a goodness fit of these models.

From the regression results of these models, the coefficient of digital economy is positive and statistically significant at the 1% level, shows that digital economy can improve creative industries development. Namely, the development of digital economy can significantly promote the development of creative industries. For every 1 percentage point increase in the development level of the digital economy, the development of creative industries can be enhanced by 0.452 percentage points.

The development of the digital economy has provided a superior environment for the development of creative industries. On the one hand, the development of Internet

**Table 3. Results of baseline regression analysis.**

| Variables | Creative industries | |
|---|---|---|
| | **(1)** | **(2)** |
| Dig | 0.642*** | 0.452*** |
| | (9.54) | (8.97) |
| ED | | 0.390*** |
| | | (4.99) |
| IS | | 0.250*** |
| | | (2.95) |
| OP | | 0.047*** |
| | | (3.76) |
| GS | | -0.023 |
| | | (-1.08) |
| Constant | 0.067*** | 0.027** |
| | (9.53) | (2.12) |
| Observations | 232 | 232 |
| R-squared | 0.980 | 0.986 |
| Province FE | YES | YES |
| Year FE | YES | YES |

Notes: Robust t-statistics in parentheses. *** p<0.01

** p<0.05

* p<0.1.FE, fixed effects.

infrastructure caused by the digital economy has promoted the cross-regional flow of creative elements, increased the speed of information dissemination and strengthened the capacity of resource integration, providing a favorable environment for the development of cultural and creative industries. On the other hand, the development of digital economy promotes the continuous accumulation of human capital elements, which promotes the continuous collision and sublimation of creative elements and provides continuous support for the development of cultural and creative industries. Finally, the development of digital economy is accompanied by the digitization and intelligence of financial industry, while the cultural and creative enterprises in most regions of China are small and medium-sized enterprises facing financing difficulties, the digitization and intelligent development of financial industry has largely relieved the financing problems of small and medium-sized enterprises and stimulated their development potential.

## 5.2 Endogenous test

Based on the above regression results, the development of digital economy contributes to the development of creative industries. Yet, the model can also suffer from endogeneity issues due to omitted variables, measurement errors, reverse causality, etc. To solve this problem, with reference to previous studies, one lag of the independent variable was used as instrumental variables. The two-stage least squares (2SLS) method was used to conduct endogeneity test.

Table 4 presents the results of the analysis of instrumental variables. In columns (1) of Table 4, we report the estimation results of the first-stage regression, The coefficient of the instrumental variable is positive and significant at the 1% level. In columns (2) of Table 4, we report the estimation results of the two-stage regression, The impact coefficient of the digital economy is positive and significant at the 1% level. The above results show that the digital economy still has a significant positive role in promoting creative industries development after considering the possible endogenous issues.

**Table 4. Instrumental variable regression results.**

|  | (1) | (2) |
|---|---|---|
|  | first stage | second stage |
| Variables | Dig | Cre |
| Dig1 | 0.9833*** |  |
|  | (0.034) |  |
| Dig |  | 0.4293*** |
|  |  | (0.049) |
| Constant | 0.0248* | -0.1260*** |
|  | (0.013) | (0.019) |
| (Kleibergen-Paap rk LM statistic) | 6.758*** | |
| Cragg–Donald Wald F-statistic | 856.464 | |
| Stock-Yogo critical values (10% maximal IV) | 16.38 | |
| Province FE | YES | YES |
| Year FE | YES | YES |
| Observations | 203 | 203 |
| R-squared | 0.993 | 0.987 |

Notes: Robust t-statistics in parentheses. *** $p<0.01$

** $p<0.05$

* $p<0.1$. FE: fixed effects.

### 5.3 Robustness test

**5.3.1 Sub-sample regression.** A certain degree of heterogeneity can develop between different regions due to the degree of development and social culture, which in turn affects the sensitivity to the regression results. In order to test the robustness of the model, this paper changed the range of samples from the whole country to the eastern, central and western regions, and conducted regression analysis separately. The results are shown in Table 5.

As can be seen from Table 5, after changing the scope of the study, the core explanatory variables are still significant after regression analyses are conducted separately for the East, Central, and West. From the results, it can be seen that the eastern region is the most significant, followed by the central and western regions, and the model fit is higher than 0.9. Therefore, it can be concluded that the model used in this paper is robust and the conclusions obtained will not change abnormally with the change of the research scope, which further proves that the development of digital economy has a significant driving effect on the development of creative industries.

**5.3.2 Delete outliers.** Data process with tail reduction for all variables in the paper at the 1% and 99% points considering any measurement inaccuracies and extreme values that may have occurred in the original data. The results are shown in column (1) of Table 6.

After the data processing, the regression results reveal that the digital economy still has a significant positive relationship with the development of creative industries, and the overall fit of the model is also basically consistent with the previous model. The results further demonstrate the regression model's reliability.

**5.3.3 Replace the dependent variable.** This paper adopts the approach of replacing the core explanatory variables for robustness testing considering that there is no uniform standard about the measurement of digital economy development index. We have changed the

**Table 5. Results of robustness check.**

| Variables | Creative industries | | | |
|---|---|---|---|---|
| | (1) | East | Central | West |
| Dig | 0.452*** | 0.459*** | 0.235** | 0.992** |
| | (8.97) | (7.88) | (2.06) | (2.17) |
| ED | 0.390*** | 0.338*** | 0.235*** | 0.651*** |
| | (4.99) | (3.19) | (3.81) | (4.76) |
| IS | 0.250*** | 0.262 | 0.057 | 0.561*** |
| | (2.95) | (1.65) | (1.07) | (3.81) |
| OP | 0.047*** | 0.068*** | 0.071 | -0.120 |
| | (3.76) | (3.00) | (1.60) | (-1.09) |
| GS | -0.023 | -0.038 | 0.030** | -0.020 |
| | (-1.08) | (-1.01) | (2.20) | (-0.65) |
| Constant | 0.027** | 0.139 | 0.048*** | -0.020 |
| | (2.12) | (0.95) | (4.27) | (-0.78) |
| Observations | 232 | 96 | 72 | 64 |
| R-squared | 0.986 | 0.985 | 0.967 | 0.967 |
| Province FE | YES | YES | YES | YES |
| Year FE | YES | YES | YES | YES |

Notes: Robust t-statistics in parentheses.

*** p<0.01

** p<0.05

* p<0.1. FE: fixed effects.

**Table 6. Results of other robustness checks.**

| Variables | Creative industries | |
|---|---|---|
| | **(1)** | **(2)** |
| Dig | 0.487*** | |
| | (4.81) | |
| DIG2 | | 0.540*** |
| | | (9.79) |
| ED | 0.275*** | 0.211*** |
| | (4.46) | (3.08) |
| IS | 0.114* | 0.237*** |
| | (1.93) | (3.45) |
| OP | 0.043*** | 0.036*** |
| | (3.72) | (3.16) |
| GS | -0.016 | -0.006 |
| | (-0.78) | (-0.33) |
| Constant | 0.038*** | 0.037*** |
| | (4.28) | (3.66) |
| Observations | 232 | 232 |
| R-squared | 0.984 | 0.988 |
| Province FE | YES | YES |
| Year FE | YES | YES |

Notes: Robust t-statistics in parentheses.

*** p<0.01

** p<0.05

* p<0.1. FE: fixed effects.

indicators in the digital economy evaluation index system. The "fiber optic cable length" was used to replace "mobile coverage", and the "software business revenue" was used to replace "e-commerce business development". In turn, the digital economy development index is recalculated and denoted as Dig2. Re-estimate the model (1) using Dig2 as the core explanatory variable. The re-estimated model results are shown in column (2) of Table 6. The re-estimated results shows that the coefficient of digital economy is still significantly positive. It is further proved that the original conclusion is stable and reliable.

## 5.4 Mediating effect of IN

On the basis of the above analysis, we conducted a mediating effects model test. Table 7 presents the empirical results of mediating effect. In column (1) of Table 7, we show the effect of digital economy on creative industries development. The estimated coefficient is positive and statistically significant at the 1% level; In column (2) of Table 7, we report the impact of digital economy on innovation efficiency. The estimated coefficient is positive and statistically significant at the 1% level, which indicates that digital economy can effectively promote innovation efficiency in China. In columns (3) of Table 7, we investigate the impact of digital economy on creative industries development by promoting innovation efficiency. The estimated coefficient of digital economy and innovation efficiency are positive and statistically significant at the 1% level. which indicates that innovation efficiency plays a partial mediating role between digital economy and creative industries development; Namely, digital economy can support the development of creative industries by promoting innovation efficiency.

**Table 7. Calculation results of mediation effect model.**

| Variables | (1) | (2) | (3) |
|---|---|---|---|
| | Cre | IN | Cre |
| Dig | 0.452*** | 0.100*** | 0.416*** |
| | (8.97) | (2.78) | (7.94) |
| IN | | | 0.360*** |
| | | | (4.32) |
| ED | 0.390*** | 0.757*** | 0.118* |
| | (4.99) | (6.38) | (1.68) |
| IS | 0.250*** | 0.673*** | 0.007 |
| | (2.95) | (4.32) | (0.11) |
| OP | 0.047*** | 0.061** | 0.025* |
| | (3.76) | (2.51) | (1.75) |
| GS | -0.023 | -0.076*** | 0.005 |
| | (-1.08) | (-2.72) | (0.21) |
| Constant | 0.027** | -0.057*** | 0.048*** |
| | (2.12) | (-2.86) | (5.38) |
| Observations | 232 | 232 | 232 |
| R-squared | 0.986 | 0.971 | 0.988 |
| Province FE | YES | YES | YES |
| Year FE | YES | YES | YES |

Notes: Robust t-statistics in parentheses.

*** $p<0.01$

** $p<0.05$

* $p<0.1$. FE: fixed effects.

# 6. Conclusions and discussion

## 6.1 Conclusions

Based on the construction of the comprehensive index system of digital economy and creative industries, this study calculated the digital economy and creative industries development indexes of 29 Chinese provinces from 2012 to 2019 using the Entropy method, and examines the impact of digital economy on creative industries development. It also investigates the mediating role of innovation efficiency between digital economy and creative industries development. and a variety of robustness tests were performed on the results. The findings are as follows: First, the results of the baseline regression indicate that the digital economy can significantly improve creative industries development. Second, the results were confirmed to be robust and reliable through a series of robustness tests. Third, the empirical results of the mediating effect model show that innovation efficiency plays a partial mediating role between digital economy and creative industries development.in other words, digital economy can promote creative industries development through the mediating mechanism of innovation efficiency.

To the best of our knowledge, this study is the first to determine whether the digital economy can promote creative industries development. Both the digital economy and the cultural and creative industries are part of the global economic transformation, and they are also the focus of academic research and government intervention [50, 51]. Treating these two economic entities separately may lead to problems [2]. Some ongoing studies have also attempted to establish the link between these two economic entities. The attention given by Duffy and

Baym to the connection between information platforms and the creative economy provides insights for the cross-development of digital research and the creative economy [52, 53]. Additionally, research in specific domains such as online gaming has laid the foundation for further cross-study of the digital economy and the creative industries [2]. The findings of this study further demonstrate, from an empirical research perspective, the impact of the digital economy on the development of the creative industries. It also explores the preliminary pathways through which the digital economy influences the creative industries, deepening our understanding of the relationship between the two economic entities. Furthermore, this study expands the analysis of both entities from theoretical and case study approaches to econometric modeling.

The following are some notable contributions. On the theoretical level, we have provided a detailed analysis of the concept and connotation of the creative industry. Furthermore, we have developed an index system for measuring the development level of the regional creative industry, which serves as a valuable reference for quantitative research in the field of creative industries. Moreover, we have made a novel attempt to incorporate innovation efficiency into the framework of digital economy and creative industry development research by exploring the impact of the digital economy on the development of the creative industry. This study provides a valuable reference for the application of digital economy and technology in cultural-related industries, expands the scope of the influence of the digital economy in industrial research, and fills the gap in existing research. In terms of practice, the creative industry plays a critical role in sustainable economic development as a green industry. This is particularly important for China, which is currently undergoing economic transformation. The shift towards a development path that emphasizes a higher proportion of cultural and creative industries in the economic system has become an important choice for sustainable green development. Against the backdrop of the digital age, our exploration of the impact mechanism of the digital economy on the creative industry provides theoretical guidance for the government in formulating industry development policies, guiding the role of the digital economy in serving industry development, and promoting the enhancement of the country's cultural soft power. This study will help the government formulate targeted policies to support the development of the creative industry and provide scientific basis for creative enterprises to proactively integrate into the digital era for their own development.

## 6.2 Limitations and future research

Some research limitations should be noted here, as is the case with any empirical work. Firstly, the sample size of this study is limited. The research conducted in this paper combines theoretical assumptions with empirical testing. Due to data availability and usability considerations, the empirical analysis was conducted using data from 2012 to 2019 at the provincial level. Future research can consider using more recent data and expand the scope of the study to the city level to explore the impact of the digital economy on the creative industries. Secondly, the article only considers the mediating role of innovation efficiency in the relationship between the digital economy and the creative industries. However, in the context of the digital age, there are multiple factors that influence the development of the creative industries. Future research can consider exploring additional potential mechanisms, such as the impact of government policies, regional economic development levels, infrastructure construction, urbanization, and other factors. This would further investigate the mechanisms through which the digital economy affects the creative industries. Additionally, the measurement indicators for the digital economy and creative industries constructed in this article are based on existing research literature and development levels. As the digital economy and creative industries

continue to evolve, their concepts and forms of expression are also constantly changing. Future research can develop a more comprehensive evaluation indicator system according to the development of the digital economy and creative industries.

### 6.3 Policy implications

Based on the aforementioned findings, digital economy has become an important path to drive the development of creative industries, and China should fully utilize the opportunities brought by digital development to promote the transformation and development of the creative industries. Therefore, this study proposes the following policy recommendations.

Firstly, improve the digital infrastructure and explore the multi-dimensional path of the digital economy to promote the development of cultural and creative industries. The government should seize the opportunity of the development of digital economy, improve the supporting facilities of digital technology, and increase the investment in high-tech research and application, so as to provide a favourable basic condition for the application of digital technology in cultural and creative industries. It will also further utilise the agglomeration of human resources, capital and innovation elements brought about by digital technology, explore the application scenarios of digital technology in cultural and creative related industries, and promote the deep integration and development of the digital economy and creative and other related industries.

The second is to improve regional innovation efficiency. The government should establish incubation bases for digital products, strengthen the application of digital technologies such as 5G, artificial intelligence, big data and cloud computing, expand the application scenarios of digital technologies, encourage innovative activities, and improve regional innovation efficiency. And it can make use of the agglomeration effect to deepen the degree of application of digital technology in creative industries, promote the digital and intelligent development of cultural and creative products, enrich the industrial form and achieve industrial upgrading, so as to promote the high-quality development of cultural and creative industries.

Thirdly, it is necessary to creating an innovative environment. Comprehensively promote the process of regional digital construction, give full play to the role of digital infrastructure in promoting resource sharing, resource integration and resource creation, optimise the environment for the development of creative industries, achieve high-quality matching of creative elements, creative talents and creative technologies, and further create a digital environment conducive to the development of creative industries.

## Supporting information

**S1 Data.**
(XLSX)

## Author Contributions

**Conceptualization:** Lei Shen.

**Data curation:** Xiaodi Zhao, Zhengyun Jiang.

**Formal analysis:** Xiaodi Zhao.

**Investigation:** Xiaodi Zhao.

**Methodology:** Xiaodi Zhao.

**Writing – original draft:** Xiaodi Zhao.

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
