## [Decision Letter · Decision Letter 0]

16 Nov 2023

PONE-D-23-30135The impact of the digital economy on creative industries development: empirical evidence based on the ChinaPLOS ONE

Dear Dr. Shen,

Thank you for submitting your manuscript to PLOS ONE. After careful consideration, we feel that it has merit but does not fully meet PLOS ONE’s publication criteria as it currently stands. Therefore, we invite you to submit a revised version of the manuscript that addresses the points raised during the review process.

We look forward to receiving your revised manuscript.

Kind regards,

JOANNA ROSAK-SZYROCKA, Assistant Professor

Academic Editor

PLOS ONE

“This research was funded by the Shanghai Municipal Financial Support Fund for Promoting the Development of Cultural and Creative Industries. [grant numbers:2022020027].”

3. PLOS requires an ORCID iD for the corresponding author in Editorial Manager on papers submitted after December 6th, 2016. Please ensure that you have an ORCID iD and that it is validated in Editorial Manager. To do this, go to ‘Update my Information’ (in the upper left-hand corner of the main menu), and click on the Fetch/Validate link next to the ORCID field. This will take you to the ORCID site and allow you to create a new iD or authenticate a pre-existing iD in Editorial Manager. Please see the following video for instructions on linking an ORCID iD to your Editorial Manager account: https://www.youtube.com/watch?v=_xcclfuvtxQ.

Reviewers' comments:

Reviewer's Responses to Questions

**Comments to the Author**

1. Is the manuscript technically sound, and do the data support the conclusions?

Reviewer #1: No

Reviewer #2: Yes

2. Has the statistical analysis been performed appropriately and rigorously? 

Reviewer #1: I Don't Know

Reviewer #2: Yes

3. Have the authors made all data underlying the findings in their manuscript fully available?

Reviewer #1: Yes

Reviewer #2: No

4. Is the manuscript presented in an intelligible fashion and written in standard English?

Reviewer #1: Yes

Reviewer #2: Yes

5. Review Comments to the Author

Reviewer #1: 1. The aim of the paper is (lines 111-114) " The aim is to investigate the relationship between digital economy and creative industries and validating the mediating effect of innovation efficiency on them. If the mediating effect of innovation efficiency can be empirically validated, then it is theoretically possible to answer the important question of how the digital

economy promote the development of creative industries. "

2. Two hypothesis have been formulated.

3. Point 6 - conclusion and discussion covers 6.1. Conclusions and 6.2. Policy implications. First there is no discussion and moreover discussion should be as separate point; it could also contain limitations and direction of future research; Conclusion should also present the application of this research or theoretical and practical implication of this research.

Reviewer #2: Dear Authors,

I have found your work as being very interesting. Few minor comments:

1. In Table 2 you have mentioned the sub-indexes you used for building digital economy index, but I think you should present more in detail the exact way you have aggregated these sub-indexes for building the digital economy index.

2. Please correlate your results with the findings of previous studies.

3. Add some limitations and directions for further research at the end of Conclusions.

6. PLOS authors have the option to publish the peer review history of their article (what does this mean?). If published, this will include your full peer review and any attached files.

Reviewer #1: No

Reviewer #2: No

---

## [Author Response · Author response to Decision Letter 0]

19 Dec 2023

Title: The impact of the digital economy on creative industries development: empirical evidence based on the China

Response to the reviewers

Dear Editor, Dear Reviewers,

Thank you for allowing a revision and resubmission of our manuscript, with an opportunity to address the reviewer’s comments.

We extend our sincere gratitude to you for your encouraging words and invaluable suggestions that have significantly contributed to the improvement of our paper.

We are uploading: (a) our point-to-point response letter to the comments, (b) a revised manuscript, (c) an updated manuscript with highlighted changes according to reviewer notes.

Please find attached the response to the queries of the reviewer and editor. We have addressed all concerns and double checked the syntax and expressions along the paper.

Best Regards, 

The authors 

Answers to questions

1. Is the manuscript technically sound, and do the data support the conclusions?

Reply: Thank you for your comments. This paper primarily investigates the impact of the digital economy on the development of the creative industries. In terms of methodology, our approach is scientifically sound and reasonable, as evidenced by the following aspects:

(1)In the paper, we proposed the research framework of the article through sufficient literature review and theoretical analysis, meanwhile, in order to select the econometric model suitable for the research topic and hypotheses, we used multiple validation. This is done as follows: firstly, the Hausmann test was conducted on the model using Stata17 software, and based on the test results, the fixed effects panel model was selected. Subsequently, time effects were taken into account in the fixed-effects model, for which the joint significance of all annual dummy variables was tested, and the results showed that the p-value was less than 0.01, so that the original hypothesis of "no time effect" was strongly rejected and the model was considered to include time effects. Therefore, this paper uses a two-way fixed effects model that considers both time and individual effects. and clustering robust standard errors to eliminate the effect of heteroskedasticity on the model. A detailed description of the steps involved in the design of the technology has been added to the paper (lines 359-367), and to facilitate your reading, we have excerpted the relevant portion from the paper as follows:

“Before estimating the model, we conducted a Hausman test using Stata 17 software to determine whether to use a random effects model or a fixed effects model in this paper. The results of the Hausman test indicated that the fixed effects model is more suitable for the research design. Subsequently, within the fixed effects model, we considered time effects. To examine the joint significance of all yearly dummy variables, we conducted a test and found that the p-value was less than 0.01. Therefore, we strongly rejected the hypothesis of "no time effects" and concluded that the model includes time effects. Then, this paper uses a two-way fixed effects model that considers both time and individual effects. and clustering robust standard errors to eliminate the effect of heteroskedasticity on the model.”

(2)The data sources of the paper are authentic and reliable, and we have provided detailed explanations and descriptions of the data source within the paper. (lines 342-356).

(3)All the conclusions in the paper are determined based on the econometric results, ensuring scientific rigor and verifiability.

2. Has the statistical analysis been performed appropriately and rigorously?

Reply: The paper conducted appropriate and rigorous statistical analysis. Prior to the statistical analysis, the model underwent F-tests and Hausman tests to select the appropriate model. In addition to the baseline regression, various methods were employed to ensure the robustness of the empirical results, including instrumental variable approach, heteroscedasticity tests, outlier removal, and variable substitution, thereby safeguarding the stability of the experimental findings.

3. Have the authors made all data underlying the findings in their manuscript fully available?

Reply: The data for this study were sourced from government public databases, and detailed descriptions of the data sources have been provided in the paper. (lines 342-356)

4. Is the manuscript presented in an intelligible fashion and written in standard English?

Reply: Yes

5. Review Comments to the Author

Reviewer #1: 

Reviewer Point P1——The aim of the paper is (lines 111-114) " The aim is to investigate the relationship between digital economy and creative industries and validating the mediating effect of innovation efficiency on them. If the mediating effect of innovation efficiency can be empirically validated, then it is theoretically possible to answer the important question of how the digital economy promote the development of creative industries. "

Reply：Thank you for your valuable suggestions. The topic of this research is the impact of digital economy on creative industries. Through theoretical analysis, we constructed a research framework and based on it, we used panel data to establish an econometric model for empirical analysis (lines 204-207; lines 312-333). The baseline regression results show that the digital economy can improve the development of creative industries. And for every 1 percentage point increase in the development level of the digital economy, the development of creative industries can be enhanced by 0.452 percentage points (lines 359-392). The results of the mediating effect model test show that innovation efficiency plays a partial mediating role between the digital economy and the development of creative industries, i.e. the digital economy can support the development of creative industries by promoting innovation efficiency (lines 442-453).

Reviewer Point P2——Two hypothesis have been formulated.

Reply：Thanks for your recognition and valuable feedback. This thesis proposes two research hypotheses based on theoretical analyses (lines 209-252), which are: 

1. Digital economy is conducive to the improvement of creative industries development. 

2. Innovation efficiency mediates the relationship between digital economy and creative industries development. 

Through empirical analysis in the article, we have validated the above two hypotheses, and the conclusion shows that the digital economy has a significant promoting effect on the development of the creative industries, while innovation efficiency plays a partial mediating role between digital economy and creative industries development.

Reviewer Point P3——Point 6 - conclusion and discussion covers 6.1. Conclusions and 6.2. Policy implications. First there is no discussion and moreover discussion should be as separate point; it could also contain limitations and direction of future research; Conclusion should also present the application of this research or theoretical and practical implication of this research.

Reply：Thank you for your valuable suggestions. We have made careful revisions to the conclusion and discussion sections. We summarize them here.

“The following are some notable contributions. On the theoretical level, we have provided a detailed analysis of the concept and connotation of the creative industry. Furthermore, we have developed an index system for measuring the development level of the regional creative industry, which serves as a valuable reference for quantitative research in the field of creative industries. Moreover, we have made a novel attempt to incorporate innovation efficiency into the framework of digital economy and creative industry development research by exploring the impact of the digital economy on the development of the creative industry. This study provides a valuable reference for the application of digital economy and technology in cultural-related industries, expands the scope of the influence of the digital economy in industrial research, and fills the gap in existing research. In terms of practice, the creative industry plays a critical role in sustainable economic development as a green industry. This is particularly important for China, which is currently undergoing economic transformation. The shift towards a development path that emphasizes a higher proportion of cultural and creative industries in the economic system has become an important choice for sustainable green development. Against the backdrop of the digital age, our exploration of the impact mechanism of the digital economy on the creative industry provides theoretical guidance for the government in formulating industry development policies, guiding the role of the digital economy in serving industry development, and promoting the enhancement of the country's cultural soft power. This study will help the government formulate targeted policies to support the development of the creative industry and provide scientific basis for creative enterprises to proactively integrate into the digital era for their own development.

6.2. Limitations and future research

Some research limitations should be noted here, as is the case with any empirical work. Firstly, the sample size of this study is limited. The research conducted in this paper combines theoretical assumptions with empirical testing. Due to data availability and usability considerations, the empirical analysis was conducted using data from 2012 to 2019 at the provincial level. Future research can consider using more recent data and expand the scope of the study to the city level to explore the impact of the digital economy on the creative industries. Secondly, the article only considers the mediating role of innovation efficiency in the relationship between the digital economy and the creative industries. However, in the context of the digital age, there are multiple factors that influence the development of the creative industries. Future research can consider exploring additional potential mechanisms, such as the impact of government policies, regional economic development levels, infrastructure construction, urbanization, and other factors. This would further investigate the mechanisms through which the digital economy affects the creative industries. Additionally, the measurement indicators for the digital economy and creative industries constructed in this article are based on existing research literature and development levels. As the digital economy and creative industries continue to evolve, their concepts and forms of expression are also constantly changing. Future research can develop a more comprehensive evaluation indicator system according to the development of the digital economy and creative industries.”

Reviewer #2:

I have found your work as being very interesting. Few minor comments

Reply：Thanks for your recognition of our work. We have made point-to-point responses to your concerns in this letter and further improved the manuscript in the revised version.

Reviewer Point P1——In Table 2 you have mentioned the sub-indexes you used for building digital economy index, but I think you should present more in detail the exact way you have aggregated these sub-indexes for building the digital economy index.

Reply：Thank you for your valuable suggestions. In this section, we mainly referred to existing research literature on the evaluation indicator system of the digital economy and the selection of sub-indexes. We also provided a detailed description of the sources and collection process of sub-indexes data. To better illustrate the process of indicator selection, we have made modifications to the description in this section. In order to facilitate your reading, we have excerpted them here.

“This paper adopted the development level of the digital economy as the core explanatory variable. The digital economy, in essence, is the result of the deep integration between the traditional economy and digital technologies [46]. It represents a series of economic activities that are based on digital technology, primarily mediated by digital platforms, and supported by digital empowerment infrastructure [47]. The measurement system of digital economy development no uniform standard, but it has been continuously studied by scholars. Referring to other scholars' research on digital economy measurement [6,9,13], we investigate the digital economy from four dimensions: Digital Infrastructure, Digital Industry Development, Digital Innovation Capability and Digital finance. At the same time, proxy variables are selected for the above dimensions. Among them, digital infrastructure refers to the digital technology-enabled software, communication facilities, computer hardware, and other components that serve as the foundation for ensuring the operation and development of the digital economy. Therefore, referring to existing research [13], digital infrastructure is represented by internet penetration rate and mobile communication coverage rate, and measured by the number of Internet users per 100 people and the number of cell phones users per 100 people. The development of the digital industry mainly considers the level of development of core digital technology industries. Draw on existing research findings [9,13,46], digital industry development is represented by internet industry output, information industry development and e-commerce business development, and measured by the telecommunications business per 10,000 people, percentage of employees in information transmission and software and information technology services, postal revenue per capital. Digital innovation capacity refers to the potential for digital technology-driven regional innovation and development, mainly represented by digital innovation element support and digital high-tech penetration, and measured by government science and technology expenditure and the degree of penetration of digital high-tech applications in listed companies. Digital finance is represented by the digital financial inclusion index (DFII), compiled by Peking University and Ant Financial jointly, which is measured by the breadth of coverage, depth of use, and digitization of digital finance [48].”

For the measurement methods and data sources of the aforementioned sub-indexes, this study provides a detailed explanation in the data description section (4.3) below. To facilitate your reading, we have included the relevant content for your review.

“The explained variable of this study are creative industries, The data comes mostly from the China Statistical Yearbook of Culture and Related Industries, the official websites of provincial governments and statistical bureaus. The explanatory variables and control variables are mainly obtained from China City Statistical Yearbook, EPS Database, China Statistical Yearbook and statistical yearbooks of each province. The digital financial inclusion index is measured using the China Digital Financial Inclusion Index released by the Digital Finance Research Center of Peking University in cooperation with Ant Financial Services Group [48]. In particular, referring to other scholars' practices, the digital high-tech penetration mainly calculated the keywords of integrated circuit, internet of Things, big data, artificial intelligence, cloud computing, block chain, ICT industry, smart city, mobile Internet, data mining, digital trade, mobile payment, satellite navigation, electronic commerce, internet plus, data visualization, virtual reality and other keywords appearing in the business scope of listed companies to reflect the penetration level of high technology, and then aggregated to the provincial scale. Due to the wide variety of indicators involved, all indicators are dimensionless in order to eliminate the influence of the indicators on the evaluation results caused by the different units of measurement. ”

Reviewer Point P2——Please correlate your results with the findings of previous studies.

Reply：Thank you for your valuable suggestions, we have added this content to the conclusion section and excerpted it below for your review.

“To the best of our knowledge, this study is the first to determine whether the digital economy can promote creative industries development. Both the digital economy and the cultural and creative industries are part of the global economic transformation, and they are also the focus of academic research and government intervention [51,52]. Treating these two economic entities separately may lead to problems [2]. Some ongoing studies have also attempted to establish the link between these two economic entities. The attention given by Duffy and Baym to the connection between information platforms and the creative economy provides insights for the cross-development of digital research and the creative economy [53,54]. Additionally, research in specific domains such as online gaming has laid the foundation for further cross-study of the digital economy and the creative industries [2]. The findings of this study further demonstrate, from an empirical research perspective, the impact of the digital economy on the development of the creative industries. It also explores the preliminary pathways through which the digital economy influences the creative industries, deepening our understanding of the relationship between the two economic entities. Furthermore, this study expands the analysis of both entities from theoretical and case study approaches to econometric modeling.”

Reviewer Point P3——Add some limitations and directions for further research at the end of Conclusions.

Reply：Thank you for your valuable suggestions, we have added this content to the conclusion and discussion section and excerpted it below for your review.

“6.2. Limitations and future research

Some research limitations should be noted here, as is the case with any empirical work. Firstly, the sample size of this study is limited. The research conducted in this paper combines theoretical assumptions with empirical testing. Due to data availability and usability considerations, the empirical analysis was conducted using data from 2012 to 2019 at the provincial level. Future research can consider using more recent data and expand the scope of the study to the city level to explore the impact of the digital economy on the creative industries. Secondly, the article only considers the mediating role of innovation efficiency in the relationship between the digital economy and the creative industries. However, in the context of the digital age, there are multiple factors that influence the development of the creative industries. Future research can consider exploring additional potential mechanisms, such as the impact of government policies, regional economic development levels, infrastructure construction, urbanization, and other factors. This would further investigate the mechanisms through which the digital economy affects the creative industries. Additionally, the measurement indicators for the digital economy and creative industries constructed in this article are based on existing research literature and development levels. As the digital economy and creative industries continue to evolve, their concepts and forms of expression are also constantly changing. Future research can develop a more comprehensive evaluation indicator system according to the development of the digital economy and creative industries.”

We are grateful for your constructive comments and suggestions, which have undoubtedly contributed to the overall improvement of our paper. We hope that the revised version meets the standards of the journal and aligns with the expectations of the readers.

---

## [Decision Letter · Decision Letter 1]

6 Feb 2024

The impact of the digital economy on creative industries development: empirical evidence based on the China

PONE-D-23-30135R1

Dear Dr. Shen,

We’re pleased to inform you that your manuscript has been judged scientifically suitable for publication and will be formally accepted for publication once it meets all outstanding technical requirements.

Kind regards,

JOANNA ROSAK-SZYROCKA, Assistant Professor

Academic Editor

PLOS ONE

Additional Editor Comments (optional):

Reviewers' comments:

Reviewer's Responses to Questions

**Comments to the Author**

1. If the authors have adequately addressed your comments raised in a previous round of review and you feel that this manuscript is now acceptable for publication, you may indicate that here to bypass the “Comments to the Author” section, enter your conflict of interest statement in the “Confidential to Editor” section, and submit your "Accept" recommendation.

Reviewer #1: All comments have been addressed

Reviewer #2: All comments have been addressed

2. Is the manuscript technically sound, and do the data support the conclusions?

Reviewer #1: Yes

Reviewer #2: Yes

3. Has the statistical analysis been performed appropriately and rigorously? 

Reviewer #1: Yes

Reviewer #2: Yes

4. Have the authors made all data underlying the findings in their manuscript fully available?

Reviewer #1: Yes

Reviewer #2: Yes

5. Is the manuscript presented in an intelligible fashion and written in standard English?

Reviewer #1: Yes

Reviewer #2: Yes

6. Review Comments to the Author

Reviewer #1: All is ok. However the article is very much rebulit. The purpose has been clearly described and this same with all methodology steps.

Reviewer #2: No further comments. The comments were properly addressed. I believe that the paper can be published in the current form.

7. PLOS authors have the option to publish the peer review history of their article (what does this mean?). If published, this will include your full peer review and any attached files.

Reviewer #1: No

Reviewer #2: No

---

## [Editor Report · Acceptance letter]

26 Feb 2024

PONE-D-23-30135R1 

PLOS ONE

Dear Dr. Shen, 

I'm pleased to inform you that your manuscript has been deemed suitable for publication in PLOS ONE. Congratulations! Your manuscript is now being handed over to our production team.

Kind regards, 

on behalf of

Dr. JOANNA ROSAK-SZYROCKA 

Academic Editor

PLOS ONE